# The impact of environmental factors on the evolution of brain size in carnivorans

M. Michaud [1✉], S. L. D. Toussaint [2] & E. Gilissen[1,3]

The reasons why some animals have developed larger brains has long been a subject of debate. Yet, it remains unclear which selective pressures may favour the encephalization and how it may act during evolution at different taxonomic scales. Here we studied the patterns and tempo of brain evolution within the order Carnivora and present large-scale comparative analysis of the effect of ecological, environmental, social, and physiological variables on relative brain size in a sample of 174 extant carnivoran species. We found a complex pattern of brain size change between carnivoran families with differences in both the rate and diversity of encephalization. Our findings suggest that during carnivorans' evolution, a trade-off have occurred between the cognitive advantages of acquiring a relatively large brain allowing to adapt to specific environments, and the metabolic costs of the brain which may constitute a disadvantage when facing the need to colonize new environments.

[1] Department of African Zoology, Royal Museum for Central Africa, Tervuren, Belgium. [2] AG Vergleichende Zoologie, Institut für Biologie, Humboldt Universität zu Berlin, Berlin, Germany. [3] Laboratory of Histology and Neuropathology, Université Libre de Bruxelles, Brussels, Belgium. ✉email: margot.michaud@mnhn.fr

Despite renewed interest in the last decades, the reasons why some animals have evolved relatively larger brains remain contentious[1]. Encephalization (*i.e.*, the relative brain size regarding the body mass) has been extensively studied within taxa whose representatives display unusually large brains, especially in mammals and notably primates[2–4], cetaceans[5–7], and birds[8–10]. Yet, the macroevolutionary mechanisms underlying patterns of brain size remain poorly understood in many taxonomic groups. Most of the studies on encephalization rely on the hypothesis that a relatively larger brain confers a selective advantage in terms of enhanced cognitive ability[11,12]. Although controversial[13,14], the association between encephalization and complex behaviours was indeed supported in several mammals[11,15] and birds[16,17]. Many studies have hypothesized that greater behavioural flexibility associated with enhanced cognition may help animals to cope with environmental challenges such as new ecological conditions[18] or facilitate the colonization of novel environments[19,20]. Similarly, an increased encephalization is argued to be favoured within species living in changing environments in order to process the overflow of information[21,22] and deal with heterogeneously distributed resources[2,23,24]. This theory, also referred as the Cognitive Buffer Hypothesis (CBH), has been significantly supported in the recent years with studies on primates[25,26], carnivorans[27], marsupials[28], teleosts[29], amphibians[30], and birds[31].

Moreover, the overall ecological complexity that a species is confronted to may also include its social environment. First proposed to explain the brain evolution in primates, the Social Brain Hypothesis (SBH) posits that cognitive requirements of social interactions are the main driver of encephalization change at the macroevolutionary scale[32]. How to characterize and accurately measure sociality is still a matter of debate[33–36]. However, several studies seem to corroborate the importance of group living in brain evolution within several mammalian groups, such as ungulates[37,38], carnivorans[37,39] and primates[37]. Additionally, certain evolutionary hypotheses also emphasize the need to consider the metabolic cost associated with increased encephalization. Indeed, the brain is among the highest demanding organs to maintain[40,41], and not only does the brain require high energetic demands, but nervous tissue growth is also very expensive[40]. This important cost of the nervous tissue in species with a larger brain could represent a selective disadvantage when resources become scarce or fluctuate over a short period of time[42–44]. The Expensive Tissue Hypothesis (ETH) therefore suggests that relatively larger brains could be selected at the expense of other energetically high demanding metabolic functions such as digestion, locomotion, or reproduction[45]. An increased number of studies have demonstrated that these hypotheses are not mutually exclusive and that multiple factors are likely to promote or constrain encephalization[46,47]. For example, despite a relatively large and energetically expensive brain, some gregarious carnivoran species that display alloparental behaviours also exhibit high reproduction rates[48]. Yet, the majority of the studies still address only one hypothesis at a time, possibly resulting in misleading interpretations[47,49]. Moreover, recent studies have stressed major differences in the evolutionary pathway to encephalization (*i.e.*, integration between brain size and body mass) for several taxonomic groups[8,50,51].

Within mammals, the taxonomic group of Carnivora have long been of particular interest in the field of neuroanatomy as representatives of both suborders (*i.e.*, Caniformia and Feliformia) display disparate patterns of encephalization[48,52–54]. Terrestrial carnivorans (*i.e* species that colonized terrestrial habitats, also called fissiped in opposition to the marine species pinnipeds, see Van Valkenburgh[55]) have succeeded to colonize almost all environment types worldwide and display an impressive diversity of lifestyles within various habitats with among other, semi-aquatic species, cursorial predators, and arboreal specialists[56,57]. In addition, terrestrial carnivorans exhibit a broad diversity of social complexity, from obligatory solitary (*e.g.*, the tiger, *Panthera tigris*) to hierarchical societies (*e.g.*, the spotted hyaena, *Crocuta crocuta*). Nevertheless, only few studies have attempted to unravel the patterns of encephalization within carnivorans (but see Gittleman, 1986[54] for a global approach as well as Lynch & Allen, 2022 who focused on environmental and dietary variables). Here we present the first comparative study investigating simultaneously the effect of multiple variables related to ecology, environment, social complexity, and physiology on encephalization in a large representative dataset of 174 species of terrestrial carnivorans (i.e., 72% of the taxonomic diversity). While controlling for phylogenetic history, we further test for body-brain allometric patterns and search for possible shifts in the carnivoran brain size evolution through time and investigated the relation between diversification rate and relative brain size. By doing so, we aim to identify which factors may have influenced the evolution of the relative size of the brain in Carnivora at different taxonomic scales, and to highlight the patterns and tempo of encephalization for this group.

Our analyses show that Carnivoran mammals display complex patterns of brain size evolution with major shifts in encephalization rate that occurred independently in several families. The home range combined with the geographic range of extant carnivoran species are highly correlated with their relative brain size compared to other environmental variables. In particular, the species geographic range is negatively correlated with their relative brain size, while the later appears to be positively influenced by the home range size. These results suggests that a trade-off between cognitive advantages and metabolic costs associated with encephalization may explain the evolution of relative brain size in carnivorans. This hypothesis allows to reconcile two widely advocated theories about the evolution of brain size within the Carnivora order considering both the metabolic cost of the neural tissue (agreeing with the "Expensive Tissue Hypothesis") and the cognitive advantage of evolving larger brain (agreeing with the "Cognitive Buffer Hypothesis").

## Results

**Brain to body allometry and patterns of encephalization in carnivorans.** We measured crania of 361 specimens belonging to 158 terrestrial carnivoran species (Supplementary Data 1). We completed the dataset with published data for 16 more species[58], for a total of 174 carnivoran species. Because all information on the ecology or metabolism are not available for every species in the literature, two different datasets were analysed. The 'dataset 1' includes all 174 terrestrial carnivoran species, and the 'dataset 2' includes the 124 species for which all information on ecology, environment, social complexity, and physiological predictors are available from literature. In addition, as we investigated two different predictors related to the social environment, we performed two separate analyses: one with the average size of the social group (referred to as 'Group size analysis'), and the other with a measure based on both the type of group-living and the hierarchical complexity (referred to as 'Social complexity analysis') (Supplementary Note 2). As expected, our PGLS analyses identified a significant positive brain to body mass correlation within Carnivora (Fig. 1). This emphasizes the major influence of the body mass on the evolution of brain size within carnivorous species. Indeed, the body mass alone explains 87% and 91% of the observed brain size variation in both datasets (dataset 1: $R^2 = 0.87$, $p < 0.001$, $\lambda = 0.56$, dataset 2: $R^2 = 0.91$, $p < 0.001$). In addition, our analyses revealed the strong influence of the

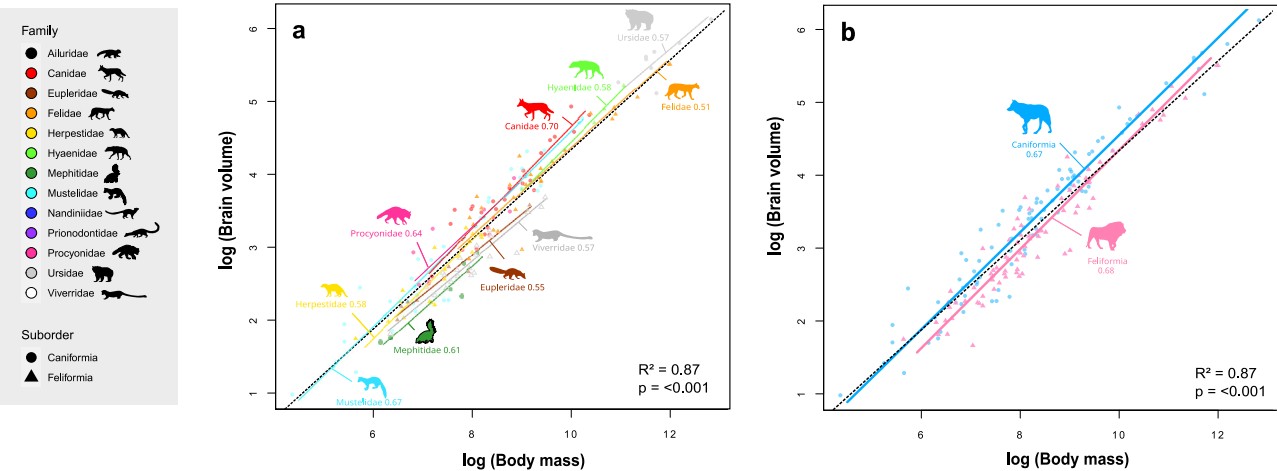

**Fig. 1 Encephalization slopes estimated for carnivoran taxonomic groups.** Slopes were estimated for **a** carnivoran families and **b** the two suborders. Values on the allometries of encephalization indicate slope characterizing the encephalization for each taxonomic group. Black doted slope represents the correlation between brain volume and body mass in terrestrial carnivorans using a phylogenetic generalized least squares regression on dataset 1 (N = 174), with R2 and p-values associated.

**Table 1 Allometric slopes estimated for (A) the carnivoran families, (B) the two suborders, and (C) the entire Carnivora order.**

| Taxonomic group | Slope | 95% CI | P-value |
|---|---|---|---|
| A- Families | | | |
| Canidae | 0.70 | 0.60–0.83 | <0.001*** |
| Eupleridae | 0.55 | 0.44–0.68 | <0.001*** |
| Felidae | 0.51 | 0.54–0.70 | <0.001** |
| Herpestidae | 0.58 | 0.54–0.85 | <0.001*** |
| Hyaenidae | 0.58 | 0.54–0.86 | 0.003** |
| Mephitidae | 0.61 | 0.40–0.91 | 0.003** |
| Mustelidae | 0.67 | 0.62–0.73 | <0.001*** |
| Procyonidae | 0.64 | 0.42–0.96 | 0.001*** |
| Ursidae | 0.57 | 0.32–1 | 0.02** |
| Viverridae | 0.57 | 0.48–0.66 | <0.001*** |
| B- Suborders | | | |
| Caniformia | 0.67 | 0.63–0.70 | <0.001*** |
| Feliformia | 0.68 | 0.64–0.72 | <0.001*** |
| C- Carnivora order | | | |
| Carnivora | 0.67 | 0.65–0.70 | <0.001*** |

Asterisks indicate the level of significance (*0.05; **0.01 and ***0.001 respectively).

phylogenetic history in carnivoran brain evolution. In particular, we found a significant phylogenetic signal of the relative brain size (datasets 1 & 2, λ = 0.99; p < 0.001), body mass (datasets 1 & 2: λ = 0.98; p < 0.001), and relative brain size (dataset 1: λ = 0.56; p < 0.001; dataset 2: λ = 0.66, p < 0.001).

Many studies have already highlighted the need to consider the allometric relationship between brain and body masses when studying the evolution of brain size[50,51]. Regarding carnivoran brain evolution, our standardised major axis method (SMA) revealed no significant differences for the allometric body-brain slope between the 13 families included in this study (Table 1A, Fig. 1a). We found the same result when considering the two suborders Caniformia and Feliformia separately (Table 1B, Fig. 1b). In a consistent way with previous studies, the slope characterizing the encephalization within all carnivorans was 0.67, meaning that body mass increases faster that brain mass (Table 1C).

The relative brain size that we quantified exhibits considerable variability within carnivoran species, ranging from 0.73 (the

African striped weasel, *Poecilogale albinucha*) to -0.85 (the spotted linsang, *Prionodon pardicolor*) (Fig. 2, Supplementary Data 2). We found significant differences in the relative brain size distribution between families (F = 9.689; p < 0.001) (Fig. 3). In particular, Ailuridae, Canidae, Ursidae, Procyonidae, Hyaenidae, Mustelidae and Felidae families display larger relative brain size whereas Herpestidae, Eupleridae, Nandiniidae, Viverridae, Mephitidae and Prionodontidae families possess smaller relative brain sizes, as expected following the standard allometric model (i.e., the body mass being the only predictor of brain size and explained the entire variation observed for the dependent variable). Analyses performed at the suborder level highlighted a significant difference of encephalization between caniformian and feliformian species (F = 30.55; p < 0.001), with caniformians having a higher relative brain size on average than feliformians (Figs. 1b, 2b).

**Tempo of encephalization**. Using a phylogenetic ridge regression approach, our results identified three significant shifts in the evolution rate of the encephalization compared to the average rate computed over the rest of the tree branches (Fig. 4). Two nodes showed a significant increase in the evolutionary rate of the relative brain size in terrestrial carnivorans: the node including all Canidae species and the node within the Mustelidae family including the Helictidinae (i.e., badgers), Guloninae (i.e., martens, fisher, tayra and wolverine), Ictonychinae (i.e., grisons and polecats), Mustelinae (i.e., weasels, ferrets, and minks) as well as Lutrinae species (i.e., otters). By using random topologies computed from the original phylogeny via tree swapping, we found that evolutionary rate shifts for relative brain size were correctly identified for these two nodes for 76% and 81% of the computed random trees respectively (Supplementary Table 1). In contrast, we identified a significant decrease in the evolutionary rate of relative brain size for the node including the Herpestidae, Hyaenidae, and Eupleridae families (i.e., Malagasy carnivorans). Nevertheless, it is worth noticing that this decrease of evolution rate was identified in only 23% of the computed random trees (Supplementary Table 1).

**Influence of predictors on the relative brain size evolution**. We investigated the influence of 13 variables classified into four different categories: ecological, environmental, social, and physiological (Supplementary Table 2). We first tested for collinearity

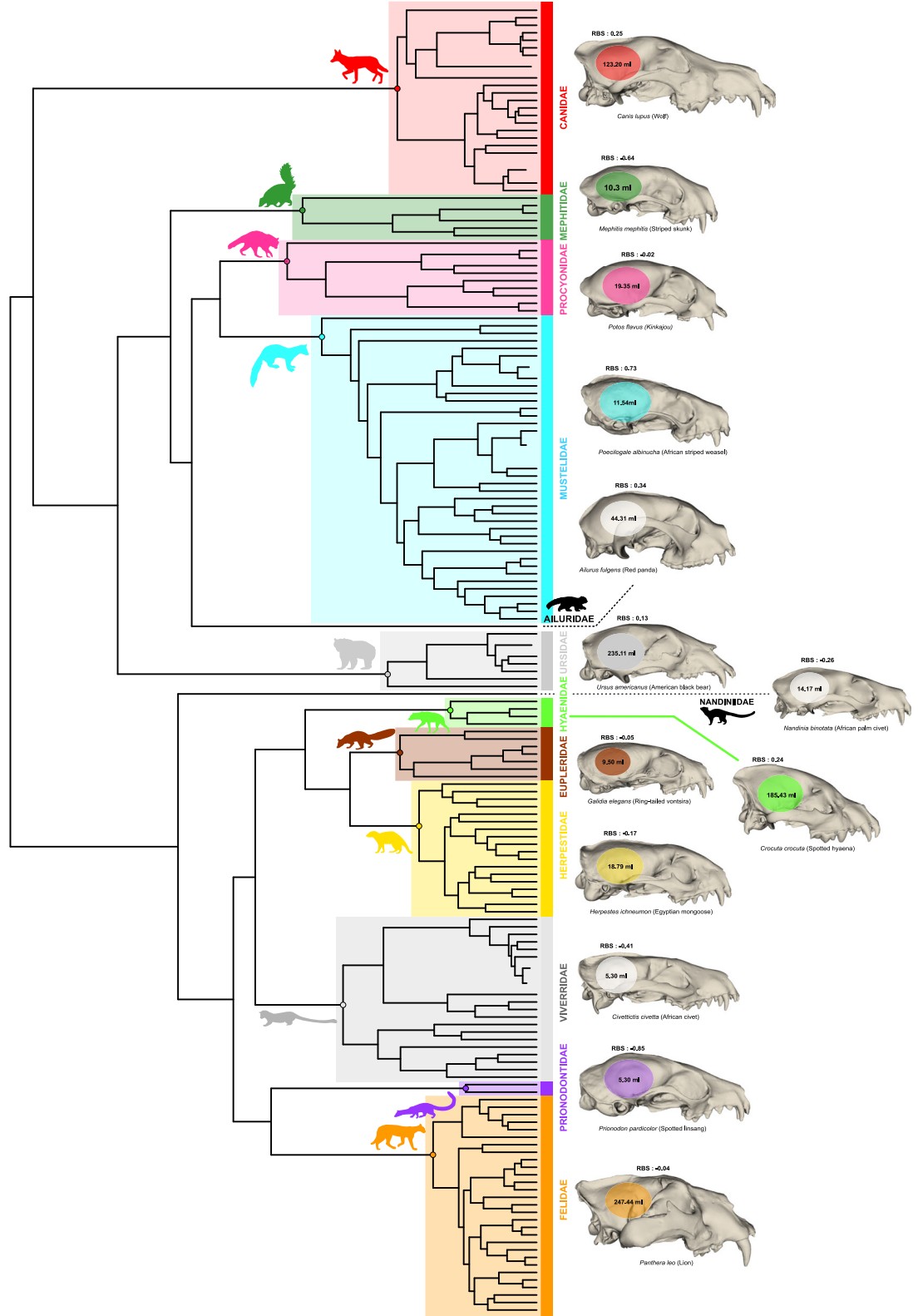

**Fig. 2 Phylogeny of the Carnivora order and illustrations of representative species.** The phylogenetic relationships of the carnivoran species used in this study, derived from Slater and Friscia (2019) associated with illustrations of a representative species of each family, its estimated brain volume, and the relative brain size (RBS). 3D crania represent different specimens from our dataset. They were digitized using a white light fringe surface scanner and a laser surface scanning. The silhouettes were drawn by the authors (M.M) and are available on the Phylopic website http://www.phylopic.org/.

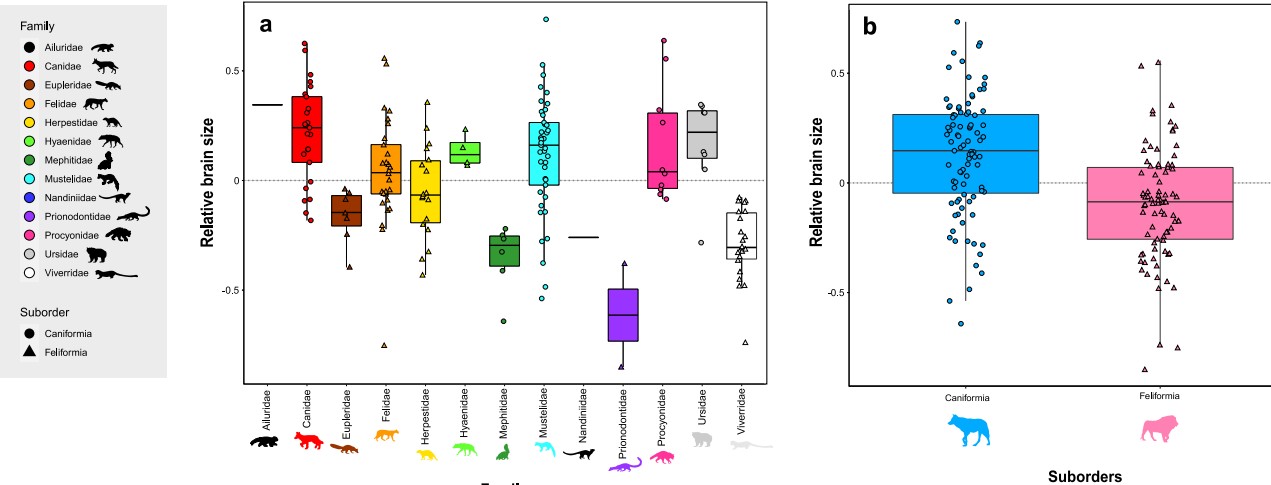

**Fig. 3 Patterns of encephalization among carnivoran taxonomic groups.** Calculated Phylogenetic relative brain size estimated for **a** families and **b** the two suborders. Grey dotted lines indicate the expected relative brain size under an allometric model (i.e., brain size is entirely explained by body mass and explained the entire variation observed for the dependent variable). Boxes represent the third and first quartiles and bold line represents the median.

between predictors using both correlation test and variance inflation factors (VIFs)[59]. Although our results showed significant correlations between some predictors (Fig. 5), VIFs results did not exceed 3.4 in all cases, meaning that all variables were more related to the response than to other predictors. PGLS analyses were performed on the dataset 2 including the 124 carnivoran species for which all information were available.

The model that best explains the evolution of encephalization in our sample of carnivoran species includes the geographic range and the home range combined for both the 'Group size' and 'Social complexity' analyses (Table 2A). This model explains 12% of the variation in the encephalization observed. While the home range is positively associated with encephalization, we found that the geographic range exhibits a significant negative correlation with the relative brain size.

The evolution of encephalization in the suborder Caniformia is best explained by a model including the geographic range, the home range, and the ability to hibernate (Table 2B, Caniformia). This model explains 28% of the encephalization variation observed within caniformian species. We found a different result for the feliformian species with only the temperature being a good predictor for encephalization in this suborder (Table 2B, Feliformia) which is negatively associated with the relative brain size. Nevertheless, this model only explains 7% of the relative brain size variation for this taxonomic group.

Our results also revealed different scenarios for the evolution of encephalization when analyses were conducted at the family level. Similarly to the analyses computed for all carnivoran species, both the geographic range and the home range are the best predictors for relative brain size evolution within Canidae species and this model explains 23% of the relative brain size variation observed (Table 2C, Canidae). For the Felidae family, the geographic range is the best predictor of encephalization, with the relative brain size being negatively correlated with it and explaining 21% of the relative brain size variation observed (Table 2C, Felidae). Finally, we found two different models explaining the relative brain size evolution within Mustelidae depending on whether the group size or the social complexity was used as social variables (Table 2C, Mustelidae). When considering the group size, the evolution of the Mustelidae relative brain size appears to be best predicted by a model including the litter size, the temperature, and the group size, which explains 49% of the relative brain size variation. When analyses were conducted with the social complexity as social predictor, we found

that the litter size and temperature best predicted the relative brain size in Mustelidae species, with 45% of the observed relative brain size variation being explained. The litter size appears to be negatively correlated with the relative brain size whereas the temperature is positively associated with the relative brain size variation in this family.

Finally, our analyses revealed no significant influence of encephalization (t = −0.48; p = 0.63) or geographic range (t = −0.68, p = 0.50) on the rate of diversification within terrestrial carnivorans. Similarly, the interaction between the relative brain size and the geographic range did not significantly impact the rate of diversification for these species (t = 0.43, p = 0.68).

## Discussion
Our findings reveal that the evolution of carnivoran brain is based on complex synergies. Brain to body allometries is remarkably constant among carnivoran taxa with all taxonomic groups being characterized by a negative allometric pattern (i.e., body mass increases faster than brain size). Ranging from 0.51 (Felidae) to 0.70 (Canidae), our results of allometric slopes are consistent with most analyses performed on other mammalian taxa[1,50,60] and within carnivoran groups[50,61,62]. Interestingly, a recent study demonstrated that within the Carnivora order, only pinniped species (i.e., aquatic taxa including the Phocidae, Otariidae, and Odobenidae) show a different brain to body allometric trajectory[50]. In order to analyse comparable ecological data, our study focuses only on terrestrial carnivorans (commonly called fissiped carnivorans, in opposition to pinniped carnivorans composed of the Phocidae, Otariidae and Odobenidae families, see Van Valkenburgh[55]). However, it should be emphasized that adaptation toward strict aquatic lifestyle may have impacted the evolution of encephalization for some carnivoran families, especially due to a significant increase in body mass.

Our analysis highlighted the fact that several shifts appeared during the evolution of the relative brain size rate in carnivorans. In particular, Canidae display a significant acceleration in the evolution of their relative brain size rate and relatively high values compared to other carnivoran families. This result of a rapid change towards higher encephalization in this group has already been stressed out by Finarelli & Flynn[62]. However, while the authors hypothesized that this evolutionary trend is the result of an increase in the social complexity within several canids lineages, our analysis show no indication that the social environment may

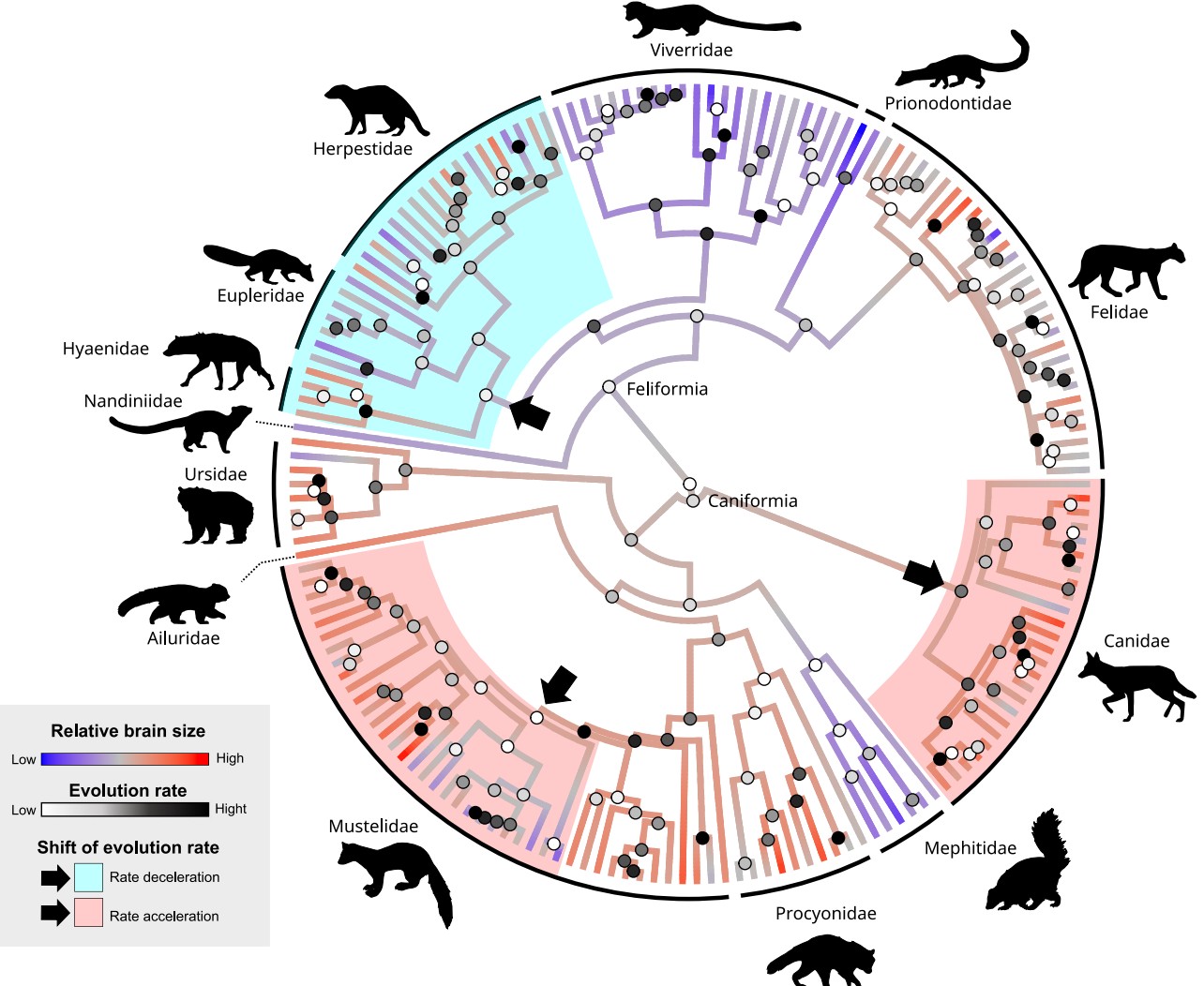

**Fig. 4 Phylogeny of the 174 species of terrestrial carnivorans studied showing the repartition of their relative brain size and the estimation of its evolutionary rate and shifts at the ancestral nodes.** Coloured dots represent the evolution rates of encephalization computed according to our phylogenetic Ridge Regression. Arrows represent the significant shifts of the evolution rate for the relative brain size. Taxonomic groups highlighted in red boxes display a significant increase in the rate of encephalization whereas the blue box indicate a significant decrease. The silhouettes were drawn by the authors (M.M) and are available on the Phylopic website http://www.phylopic.org/.

have played a role during the evolution of the relative brain size in this group. On the contrary, our results demonstrates that environmental parameters, in particular the home range size and the geographical distribution of species have likely influenced the evolution of the relative brain size within canids. Finarelli & Flynn[62] also observed a significant shift in the encephalization rate within Ursidae which we did not reproduced in our analyses. This difference could be explained by the fact that the increase in relative brain size in the Ursidae family was probably a relatively slow process rather than an abrupt change, a phenomenon that we cannot detect with the statistical approaches used in our study.

Within caniformian species, Mustelidae also display a singular pattern. This group has been characterized by a significant shift of relative brain size rate resulting in a broad disparity of relative brain size. This relative brain size diversity is certainly related to the impressive morphological diversification that mustelids undergo during their evolutionary history. Indeed, although the precise number of taxonomic bursts is still debated, several authors have hypotheses that the unique ecological and morphological diversity observed within mustelids species[63–66] are the result of an adaptive radiation event[67,68]. Moreover, some

mustelids species (in particular the node including the Helictindinae, Guloninae, Ictonychinae, Mustelinae and Lutrinae subfamilies) experienced a diversification in their body shape during the Mid-Miocene Climate Transition which coincides with the significant change in the rate of evolution of the relative brain size revealed by our analyses[69]. Surprisingly, the shift in diversification rate for the skull shape identified by Law[69] appears to be a subsequent event to the diversification of the relative brain size revealed by our analyses, meaning that cranial shape did not constrain the relative brain evolution for these species. Thus, it is possible that the rapid evolution of encephalization in this group, in addition to changes of overall body shape and an evolutionary shift toward more elongated bodies[70], have played a key role in the evolutionary history of Mustelidae, allowing these species to diversify rapidly into new ecological niches.

The Eupleridae family is a unique case within carnivorans, all species being endemic to the Madagascar island. Geographical isolation is often considered as a special case with regard to phenotypic evolution[71–75]. As for the brain, recent evidence within bird taxa suggests that there is a predictable evolutionary trajectory toward larger encephalization for species evolving

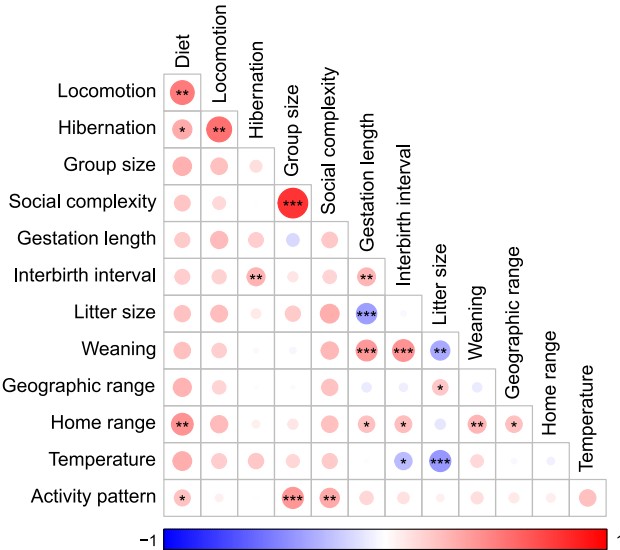

**Fig. 5 Correlation between the different predictors used in this study.** Correlation matrix of the ecological, environmental, social, and life-trait history predictors tested in this study with associated values and *p*-values. Negatively correlated variables are shown in blue while those positively correlated are coloured in red. The statistical significance of each correlation is signified by asterisks whereas size of the circle is defined by the correlation strength. Asterisks indicate the level of significance (*0.05; **0.01 and ***0.001 respectively).

within insular systems compared to mainland populations[76]. Common ancestors of Malagasy carnivorans have arrived in the island by a single event of colonisation, approximately 18-24 million years ago[77]. Yet, despite the fact that Eupleridae species evolved in an insular system without any competition, our analyses did not recover a shift in the rate of relative brain size evolution for this group. In addition, Eupleridae species display relatively low relative brain size compared to other carnivoran families. Together these results suggest that the increasing encephalization hypothesized for island species does not apply for all taxonomic group and corroborates recent research suggesting that brain evolution within mammalian insular populations is a phenomenon much more complex than previously assumed[78,79].

Encephalisation in carnivoran species is positively associated with the average size of the home range. Consistently with previous studies on mammalian taxa[80–84], this result is in adequation with the Cognitive Buffer Hypothesis (CBH) and could be explained by the fact that a larger brain may constitute a selective advantage for species that need to memorize a large amount of temporal and geographic information[37,85]. Indeed, the ability to use mental representation of the surrounding environment and to memorize the food distribution is very important as it impacts the foraging success[86–88]. Interestingly, Swanson *et al*[46]. demonstrated that the parietal cortex size, more specifically the somatosensory cortex involved in spatial reasoning, is significantly correlated with the home range size within carnivoran species. However, the authors did not find any correlation between the relative brain size and the home range and suggested a possible evolutionary trade-off between the size of the frontal cortex and the parietal cortex which may limit the energetic costs associated with the increase of neuronal tissues. However, this study was based on a relatively small sample size (i.e., 36 species) due to technical difficulties in acquiring images from CT-scanning. We therefore suggest that this restricted sample size was possibly not sufficient to highlight the correlation that we and other studies[82,84] found between the relative brain size and the home

**Table 2 PGLS models selected by BIC comparison that best predict the evolution of encephalization for (A) the order Carnivora, (B) the suborders Caniformia and Feliformia, and for (C) the Canidae, Felidae and Mustelidae families.**

A- "Group size" & "Social complexity" analyses (same results)
Carnivora (N = 124)

| Predictor | SE | t | p |
|---|---|---|---|
| Intercept | 0.19 | 3.31 | 0.001** |
| Geographic range | 0.01 | −3.95 | <0.001*** |
| Home range | 0.01 | 2.52 | 0.013* |

Model summary: λ = 0.73, R² = 0.118, p < 0.001***

B- "Group size" & "Social complexity" analyses (same results)
Caniformia suborder (N = 71)

| Predictor | SE | t | p |
|---|---|---|---|
| Intercept | 0.23 | 2.15 | 0.035* |
| Geographic range | 0.01 | −2.81 | 0.006** |
| Home range | 0.01 | 3.43 | 0.001** |
| Hibernation (no) | 0.07 | 3.64 | <0.001*** |

Model summary: λ = 0, R² = 0.28, p < 0.001***

B-"Group size" & "Social complexity" analyses (same results)
Feliformia suborder (N = 53)

| Predictor | SE | t | p |
|---|---|---|---|
| Intercept | 1 | 2.16 | 0.035* |
| Temperature | 0.003 | −2.30 | 0.026* |

Model summary: λ = 0.85, R² = 0.07, p = 0.03*

C-"Group size" & "Social complexity" analyses (same results)
Canidae family (N = 23)

| Predictor | SE | t | p |
|---|---|---|---|
| Intercept | 0.25 | 3.78 | 0.001** |
| Geographic range | 0.02 | −3.21 | 0.004** |
| Home range | 0.02 | 2.49 | 0.021* |

Model summary: λ = 0, R² = 0.32, p = 0.008**

C-"Group size" & "Social complexity" analyses (same results)
Felidae family (N = 24)

| Predictor | SE | t | p |
|---|---|---|---|
| Intercept | 0.52 | 2.83 | 0.0097** |
| Geographic range | 0.03 | −2.69 | 0.013* |

Model summary: λ = 0.94, R² = 0.21, p = 0.01**

C-"Group size" analysis
Mustelidae family (N = 27)

| Predictor | SE | t | p |
|---|---|---|---|
| Intercept | 0.78 | −2.55 | 0.018* |
| Litter size | 0.01 | −4.13 | <0.001*** |
| Temperature | 0.003 | 3.13 | 0.005** |
| Group size | 0.007 | −1.75 | 0.0936536. |

Model summary: λ = 1, R² = 0.49, p < 0.001***

C-"Social complexity" analysis
Mustelidae family (N = 27)

| Predictor | SE | t | p |
|---|---|---|---|
| Intercept | 0.79 | −2.97 | 0.007** |
| Litter size | 0.02 | −3.72 | 0.001** |
| Temperature | 0.003 | 3.48 | 0.002** |

Model summary: λ = 1, R² = 0.45, p < 0.001***

For each taxonomic group, we conducted two separated analyses with either the "Social complexity" or the "Group size" as social predictors. Asterisks indicate the level of significance of *p*-values (*<0.05; **<0.01 and ***<0.001 respectively).

range. In the future, additional research on the endocranial shape will be necessary to assess whether the increase in the parietal cortex for species with a large home range may be responsible for the increase in the overall size of the brain.

Unexpectedly, our analyses revealed that the encephalization in Carnivora is negatively associated with the geographic range, suggesting that species with a large geographic distribution would tend to have proportionally lower encephalization. This also suggests that species with a low encephalization might better succeed in colonizing new habitats and settling durably in different environments. This result is surprising as numerous studies have provided

examples of the opposite trend, with large-brained species being the most successful at colonizing new environments[19,20,76,89,90]. In this context, the Cognitive Buffer Hypothesis (CBH) is again often used to explain how relatively larger brain facilitates behavioural buffering under unpredictable environmental changes[91–93]. However, it is important to consider that even if some research investigated the relation between brain evolution and environmental changes within mammals[19,28], the majority of studies focused on bird taxa[18,31,76,89,93]. In addition, studies have also demonstrated that environmental factors such as scarcity of resources related to high seasonality could promote smaller brain that are therefore less energetically expensive[28,43,44,94]. In this case, the Expensive Tissue Hypothesis (ETH) is preferred as the major constraint for brain mass increases. Similarly, Fristoe and Botero[95] provided evidence that some bird taxa facing difficult environmental conditions display relatively smaller brain, which would allow them to allocate more energy to other important functions (*e.g.*, reproduction and digestion). It is therefore likely that different encephalization strategies are possible in the same taxonomic group and are conditioned primarily by the environment.

As the metabolic cost of the brain is among the highest of any organ to maintain[45,96,97], we suggest that having a relatively small brain could confer a metabolic advantage to adapt in environments where resources are scarce or vary over time. A brain that requires less energy would therefore allow carnivorans to settle more easily and sustainably in new environments. Our results further support this hypothesis and demonstrate that diversification rates are not explained by the relative brain size nor geographic range within terrestrial carnivorans, implying that the geographic distribution of carnivoran species is not correlated to their phylogenetic age, and that species with a higher relative brain size would not tend to diversify more than smaller brained close relatives. Our hypothesis also echoes a recent study which demonstrates that home range size in terrestrial mammals is the result of a trade-off between the ecological specialization of a species and its ability to move, while the geographic distribution is linked to the ability of a species to resist to various environments[98]. Like Huang *et al.*[98], our result reflects a potential impact of natural selection at two different evolutionary scales: the use of the surrounding environnement by individuals, and the dispersal of species within different habitats and localities. Consistently with a study conducted on catarrhine primates[25], this also would mean that both CBH and EBH have a crucial role for the brain evolution in carnivoran species but act at different evolutionary scales. Thus, the relative brain size could be seen as the results of a trade-off between the cognitive advantages of having a large brain, allowing species to exploit complex habitats, and the metabolic cost of having a larger brain, which become a disadvantage when colonizing new environments or facing environmental changes is needed. This scenario is also supported by studies which show that the risk of extinction is higher in mammals with relatively larger brains[99]. As the surrounding environment of a species is characterized, among other, by the temperature, it is interesting to find in our analysis a significant impact of the average temperature on the relative brain size for the Feliform and Mustelidae species. It should be emphasized that the average temperature represents only one aspect of what characterizes the abiotic environment complexity, yet, a recent study of Lynch & Allen (2022)[100] which focused on some specific environmental and diet parameters did not find any influence of the temperature range on carnivoran brain size evolution when multiple variables were analysed together. However, it is interesting to note that their study revealed a significant effect of vegetation index on brain size evolution, an environmental variable that could be related to the geographic range of species. Interestingly, although the percentage of meat appears to be significantly influential when only this variable is considered for the Felidae, Mustelidae and Procyonidae families (i.e., PGLS with

only 'percent of meat in diet' as single predictor), the authors also did not find any impact of diet variation or quality on brain size evolution when several predictors were considered alongside, which is consistent with our own results. In the future, the inclusion of new environmental data related to abiotic parameters, would be important to investigate in parallel with the geographic range and the vegetation index in order to draw finer conclusions on the impact of this specific environmental aspect on the evolution of the mammalian brain. Finally, it is possible that populations with a large geographic distribution may also display higher intra-specific variability of their relative brain size[98]. Our analyses do not allow us to specifically test this phenomenon, but we acknowledge the importance of investigating intra-specific brain size differences in the future to better understand the evolution of encephalization at different scales and test the strength of our hypothesis.

Our study overall highlights the importance of the taxon-level effect on the study of encephalization. Indeed, our results showed different evolutionary trends depending on families which nevertheless belong to the same taxonomic order. While the evolution of the relative brain size within Canidae is highly influenced by the same ecological factors as when considering the entire Carnivora order (i.e., the home range and the geographic range), the evolution of the relative brain size in Felidae seems to be solely impacted by the geographic range. In addition, the Mustelidae family exhibits a very different pattern compared to other carnivoran families, their relative brain size being significantly influenced by reproductive (i.e., litter size), environmental (i.e., average temperature) and social (i.e., average group size) constraints. This observation is also true for the study carried out by Lynch & Allen (2022) which highlights the influence of distinct environmental and diet variables depending to the different families. It is therefore possible that selective pressures do not influence all taxonomic groups in the same way, even if these taxa share a common evolutionary history[101]. Although the taxonomic level of the order appears to be a significant phylogenetic grouping in terms of gyrification evolution in mammals[102], it is also possible that some evolutionary trends are only notable at a certain taxonomic level. For example, Finarelli[103] showed that hierarchical effects were evident in the relationship between encephalization and reproductive features within carnivorans. Similarly, the study of hyaenids endocranium have revealed a significant correlation between the frontal cortex size and the sociability in this family[104] but this trend has not been found back when all Carnivora species were considered together[46]. This suggests that some encephalization patterns might be noticeable in large-scale taxonomic groups, whereas some patterns may become apparent only when investigating lower taxonomic levels such as families. One explanation may rely in the fact that variables thought to influence the relative brain size at macroevolutionary scale are not distributed identically between the different taxonomic groups or between different taxonomic scales (see Supplementary Data 3). The most telling example is probably the ability to hibernate, which is observed only in species belonging to the Caniformian suborder and which appears to be a significant factor in the evolution of the relative brain size for this suborder, but this factor is not found to be significant at the family level. Such differences in distribution hold true for continuous variables. For example, we found significant differences of variance for the geographic range between the taxonomic groups studied, with Canidae having a broader variance than all others. Similarly, although there is no significant difference in the variance, the home range appears as significantly larger for the Felidae family than for all the other taxonomic groups. There are therefore fundamental differences in the distribution of the predictive variables according to taxonomic groups that may explain the results we observe in our study. We

suggest that it is essential to consider this aspect in future studies related to brain evolution.

Altogether, these results reveal that carnivoran brain size evolution have been impacted by environmental factors during their evolutionary history, more specifically by home range and geographic range. This study provided new insights into the complex history of relative brain evolution for this iconic mammalian group and highlight the importance of carefully considering taxonomic frameworks in large-scale comparative studies. In the future, finer analysis of endocranial structures made possible by the technological development of recent decades could be decisive in bringing new elements to our understanding of mammalian brain evolution.

## Methods

**Materials**. The number of specimens analysed ranged from one to eleven individuals for each species (Supplementary Data 1). The specimens sampled were adults, preferentially wild caught and both males and females were included when possible. Crania were digitized using a white light fringe surface scanner (Breuckmann StereoSCAN3D model with a camera resolution of 5 megapixels) at the plateform 'Plateau de morphométrie' (UMS CNRS 2700 OMSI, Muséum National d'Histoire Naturelle, Paris, France) and using a laser surface scanning (FARO EDGE Scan Arm HD) at the Natural History Museum (London, UK). Some 3D crania models were also acquired from online database collections (Morphosource, Digimorph, Phenome10k and African fossils).

**Brain volume estimation**. We estimated the brain volume of each specimen using a method specific to extant carnivoran species based on three external measurements of the cranium[58], more specifically the length, height, and width of the braincase (Supplementary Fig. 1). This model-averaging technique allows to correctly predict the endocranial volume of carnivoran species with more than 98% accuracy and have been widely used as a proxy for the brain volume in many previous research on mammalian brain evolution[46,62,105–109]. Although studies have pointed out some limitations for models based on external skull measurements, these issues appear to be specific to intraspecific scale[110,111]. These measurements were collected from the surfaced 3D images using the Idav Landmark software. All cranial measurements and result of brain volume estimation, also referred as brain size, are provided in the Supplementary Data 1. The corresponding averaged brain volume was then calculated for each species (Supplementary Data 2). The pooled standard deviation of log (brain size) for species with at least three specimens was only 0.06, suggesting that intraspecific variation is relatively low. Regarding the published brain size data, a common pitfall in comparative studies is the bias introduced by using brain volume data extracted from different publications[33]. Thus, we performed a pairwise test with Bonferroni correction to compare between the brain volume extracted from the literature and our estimations in order to test for the inter-operator bias. We found no significant differences between the brain volume in the literature and our results (p = 0.99) and therefore included these 16 species in our sampling.

**Selection of predictors**. For phylogenetic comparative analyses, the number of variables should be at least 10 times the number of predictors[112]. Therefore, we chose 13 variables (with 12 variables investigated simultaneously) considered as good candidates to predict the evolution of the brain size in mammals, classified into four different categories: ecological, environmental, social, and related to metabolism and reproduction (referred as physiological predictors). Description for each variable is available in Supplementary Note 1 and coding for each species are presented in Supplementary Table 2. Continuous variables were computed as the mean values of both males and females when available. Ecological variables include diet, locomotion, activity pattern and average home range size. Environmental variables include the average geographic range (i.e., the geographic distribution of a species in km²) as well as the average temperature measured in this geographic range. The geographic range itself does not represent an environmental parameter that can influence fitness. Yet, the geographic distribution is a relevant proxy for the ability of a species to immigrate, survive and thrive in different environments, and traits which influence the geographic distribution are known to evolve under natural selection. Therefore, in our study we used the geographic range as a proxy of species abilities to disperse and colonize new geographic landscapes. Physiological predictors include the ability to hibernate, the gestation length, the average weaning time, the average interval between each litter and the average litter size. Because all information on the ecology or metabolism are not available for every species, two different datasets were analysed. The 'dataset 1' is a complete dataset including all 174 terrestrial carnivoran species, whereas the 'dataset 2' includes the 124 species for which all information on ecology, environment, social complexity, and physiological predictors are available. Information for body mass and all predictors were extracted from the PanTHERIA online database[113], the *Handbook of the Mammals of the World*[57] and supplemented by published sources for missing data[114–129]. Finally, in order to test for the Social Brain

Hypothesis (SBH) in carnivoran species, we included variables reflecting social complexity. One of the most widely used proxy of social environment is the mean group size. However, some authors have suggested that a number of individuals do not reflect accurately the complexity of social interactions within a group[33–35,130]. We therefore performed two separate analyses: a first analysis with the average size of the social group (referred to as 'Group size analysis'), and a second analysis with a measure based on both the type of group-living (e.g., pair-living, familial group, extended group) and the hierarchical complexity within groups (referred to as 'Social complexity analysis') (Supplementary Note 2). Because we assume that the variability of these predictors may be different depending on taxonomic scales, we estimated the variance of each predictor for the entire order Carnivora, the suborders Caniformia and Feliformia, as well as families with more than 20 species in our sample (i.e., Canidae, Felidae and Mustelidae families). For the categorical predictors, we used a chi-squared contingency table to test for proportions homogeneity of the different categories according to the taxonomic groups. Similarly, we performed Bartlett's test to assess the homogeneity of variances for continuous predictors at different taxonomic scales in parallel with ANOVA tests which provides information on the differences in means between these groups. The detailed results and associated figures of these analyses are available in the Supplementary Data 3.

**Testing for differences in allometric patterns**. All statistical analyses were conducted with R version 4.0.2. (R Core Team, 2020). In order to assess the statistical differences in allometric patterns of relative brain size and body mass covariation between taxonomic groups, we performed a standardised major axis method (SMA)[131] using the dataset 1. Both brain mass and body mass values were log10 transformed prior to calculation of the regression slope.

**Phylogenetic relative brain size**. To take into account the effect of phylogenetic non-independence, we used Phylogenetic Generalized Least-Squares analyses (PGLS) and the recent time-scaled phylogeny of the Carnivora order proposed by Slater and Friscia[132]. Brain size and body mass values were log10 transformed prior to statistical analysis. We computed a PGLS analysis between the brain and body masses on the dataset 1 in order to obtain the phylogenetic relative brain size used for further analyses (i.e., the residuals of the PGLS). Significant differences between taxonomic groups were assessed using ANOVA analyses.

**Tempo of encephalization**. In order to investigate rate and patterns of brain size evolution within carnivoran mammals, we used a phylogenetic ridge regression approach available under the *RRphylo* package[133] using the *search.shift* function. This Phylogenetic Comparative Method (PCM) computes the rates of encephalization evolution (i.e. the magnitude of relative brain size changes relatively to the time unit) for each branch of the topology and returns the ancestral state estimated at each node without assuming any evolutionary model prior to analyses. We then searched for significant shifts in the evolutionary rate of encephalization across the phylogenetic tree. The significance of evolutionary shifts was tested through randomization.

To assess the effect of phylogenetic uncertainty in the estimation of relative brain size rate evolution and on the shifts calculation, we used the *overfitRR* function implemented in the *RRphylo* package[133]. Based on permutation process, this method produces alternative phylogenies with different number of tips, tree topology and branch lengths. For each alternative phylogeny, the function estimates significant shifts in the evolution rate and compares the results to the original phylogeny. This process is repeated 100 times and the percentage of significant results is returned for each shifts detected.

**Impact of the relative brain size on diversification rates**. Studies have shown that species with larger relative brain size tend to display higher diversification rates compared to closed relatives[134–136]. Under this scenario, species with a large geographic range would therefore not necessarily be better in their ability to migrate and settle sustainably in new environments, but rather may simply be older than small-ranged species. Similarly, it would be possible to advocate that relatively large-brained species could be better able to colonize new habitats, resulting in a taxonomic diversification that can give rise to a large number of small-ranged new species. In order to test this hypothesis, we investigated for the relationship between the diversification rate and the relative brain size as well as the effect on geographic range at the species level in our dataset. To do so, we first estimated diversification rates for each species using the function *evol.distinct* available under the R package *picante*[137]. Then, we used PGLS analyses with diversifications rate as the response variable while the relative brain size, the geographic range, and the interaction between these two parameters were used as predictors.

**Influence of predictors on the evolution of encephalization**. We investigated the influence of our 13 predictors on the encephalization in a phylogenetical framework using the dataset 2. To do so, we computed PGLS analyses. For each PGLS model, a phylogenetic signal was calculated using the Pagel's lambda (λ) estimated by maximum likelihood[138]. Continuous variables were log10 transformed prior to statistical analysis to satisfy the assumption of normality. We checked the distribution of all transformed variables to make sure that they were symmetrically distributed. We first tested for collinearity between predictors using both correlation test and Variance

inflation factors (VIFs)[59]. As expected, we found strong collinearity between the average group size and the social complexity. However, this was not an issue as we kept these two analyses separated. Finally, all model diagnostic plots (e.g., outlier analysis, Q-Q plots) were inspected to make sure that assumptions were not violated[59]. We created a new method based on incrementation process to obtain parsimonious predictive models. We used an iterative selection strategy by adding and testing variables (the 13 predictors) one by one to select the combination of variables that increased the most the explained variance. Model comparisons were conducted using BIC rather than AIC because the former used higher penalization for additional terms commonly recommended for complex models. These analyses were carried out at different taxonomic scales, considering the entire Carnivora order (N = 124 species), the suborders Caniformia (N = 71 species) and Feliformia (N = 53 species), as well as families with more than 20 species sampled (i.e., Canidae N = 23; Felidae N = 24 and Mustelidae N = 27). In addition, to better describe the impact of each variable on the relative brain size within carnivoran, we performed PGLS analyses on each variable separately. The results of these additional analyses are presented in the Supplementary Table 3.

**Statistics and Reproducibility**. Brain volume estimation was based on three external measurements of the cranium[58] (see Supplementary Fig. 1). All cranial measurements and brain volume estimations are provided in the Supplementary Data 1 to ensure the reproducibility. We tested for intraspecific variation using a pooled standard deviation of log (brain size) for species with at least three specimens and found low variation (0.06 standard deviation). In addition, we tested for inter-operator bias using pairwise test with Bonferroni correction to compare between the brain volume extracted from the literature and our estimations and found no significant differences between the brain volume in the literature and our results ($p = 0.99$).

All statistical analyses were conducted with R version 4.0.2. (R Core Team, 2020). We analysed 13 variables considered as good candidates to predict the evolution of the brain size in mammals. Continuous variables were computed as the mean values of both males and females when available. The 'dataset 1' is a complete dataset including all 174 terrestrial carnivoran species, whereas the 'dataset 2' includes the 124 species for which all information on ecology, environment, social complexity, and physiological predictors are available. Two set of analyses was performed depending on the sociable variable: a first analysis with the average size of the social group ('Group size analysis'), and a second analysis with a measure based on both the type of group-living and the hierarchical complexity within groups ('Social complexity analysis') (see Supplementary Note 2). A chi-squared contingency table was used for categorical predictors to test for proportions homogeneity of the different categories according to the taxonomic groups. Likewise, we performed Bartlett's test in parallel with ANOVA tests to assess the homogeneity of variances for continuous predictors at different taxonomic (see Supplementary Data 3).

Differences in allometric patterns were assess through standardised major axis method (SMA)[131] using the dataset 1. We obtain the phylogenetic relative brain size by computing a PGLS analysis between the brain and body masses on the dataset 1 and significant differences between taxa were assessed using ANOVA analyses. We used a phylogenetic ridge regression approach available under the *RRphylo* package[133] to investigate rate and patterns of brain size evolution. We then used we the *overfitRR* function implemented in the *RRPhylo* package[133] to assess the effect of phylogenetic uncertainty in the estimation of relative brain size rate evolution and identified shifts calculation on the topology. We assess for the relationship between diversification rate and relative brain size by analysing diversification rates for each species through the *evol.distinct* function available under the R package *picante*[137]. We then used PGLS analyses while relative brain size, geographic range, and the interaction used as predictors.

Finally, we investigated the influence of the 13 predictors on the encephalization in a phylogenetical framework using the dataset 2 using PGLS analyses. Continuous variables were log10 transformed and phylogenetic signal was calculated using the Pagel's lambda (λ) estimated by maximum likelihood[138]. We checked the distribution of all transformed variables and tested for collinearity between predictors using correlation test and Variance inflation factors (VIFs)[59] before running all model diagnostic plots (e.g., outlier analysis, Q-Q plots) to make sure that assumptions were not violated[59]. Finally, model comparisons were conducted using BIC rather than AIC as the former used higher penalization for additional terms.

**Reporting Summary**. Further information on research design is available in the Nature Research Reporting Summary linked to this article.

## Data availability

## Code availability

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

## Acknowledgements

We thank the Fyssen Foundation for funding this research. We thank the curators and staff of the following museums for the loan of the specimens: G. Veron; A. Verguin, C. Bens, Muséum National d'Histoire Naturelle, Paris; R. Portela Miguez, Natural History Museum, London; L. Costeur, Naturhistorisches Museum Basel, Basel; J.M. Chupasko, Museum of Comparative Zoology, Harvard; S. Peurach, National Museum of Natural History, Washington. Specimens from DigiMorph and Morphosource were provided by B. Figueirido, P. Owen, T. Rowe, J. Theodor, J. Tseng and B. Van Valkenburgh. Scans of specimens from the Muséum National d'Histoire Naturelle and the Royal Museum for Central Africa have been acquired on the "Plateau de morphométrie," Muséum National d'Histoire Naturelle, UMS CNRS MNHN 2700 "Outils et Méthodes de la Systématique intégrative." We are very grateful to A.C. Fabre for providing several carnivoran cranium scanners used for this study. We are grateful to F. Machado for his useful comments and his valuable help on intraspecific variation estimation. We also thank D. Tamagnini for providing helpful comments during the writing of our manuscript. Finally, we would like to express our thanks the two anonymous reviewers who helped improve our manuscript.

## Author contributions

M.M. contributed to conception of the study, carried out the acquisition of data and performed the statistical analyses with input from S.L.D. M.M., and S.L.D., wrote the main manuscript with inputs from E.G. All authors revised and approved the manuscript.

## Competing interests

The authors declare no competing interests.
