## [Peer Review File · Communications Biology]

Reviewers' comments:

Reviewer #1 (Remarks to the Author):

This study focus on brain size evolution in Order Carnivora. The authors have compiled an impressive dataset of 174 species and show how the rate of brain size evolution have changed in different places of the phylogeny of carnivores. In addition, the authors also test which traits might be more influential for the evolution of brain size. The study is important and interesting, and the figures are very well presented (the reconstructed skulls are super nice!). However, I have several comments and concerns that might need to be addressed prior to publication.

A first general comment is the inclusion of certain variables in the analysis. This is a bit of a philosophical debate, but I am not comfortable with the inclusion of geographical range as a predictor of brain size. All the independent variables (predictors) included are either species traits (life-history, sociality, etc.) that could influence the fitness of other traits (here brain size) or abiotic/environmental variables (i.e., temperature) that one could also argue that can change the fitness of other traits (i.e., in this case brain size). However, geographical range is neither a "trait" nor a proper environmental variable that can be a selective agent (but maybe a proxy for some other variable?). In this sense, individuals of a species do not possess a "geographical range" value that could be inherited or selected. I can see why geographical range could be useful in other species comparison contexts (it can be a predictor of extinction risk or maybe a response of certain traits), but I do not see how it could affect brain size evolution. So, my suggestion would be to either exclude it from the analysis, or instead, explain which is the prediction of including this variable.

Related to the previous comment, the significant effect of Geographical range is very difficult to interpret, as it could mean different things. For instance, the fact that large-brained species have smaller ranges is interpreted as a lower ability to colonize new habitats (Line 363). However, it might be the case that large-brained species are better able to colonize new habitats, and then speciate, giving rise to a lot of small-ranged species. Indeed, some previous studies have found that large-brained mammals and birds tend to speciate more often (Sol et al. 2005; Sayol et al. 2019; Creighton et al. 2021). Therefore, if geographical range is maintained in the analyses, I think that a more nuanced discussion could help to interpret its significance.

Regarding the inclusion of temperature. I guess the main reason is that previous works have found that environmental variation is a good predictor of relative brain size. However, note that it is "environmental variation" what matters and not temperature per se, so it might be interesting to include other environmental variables that directly measure variation such as amplitude or SD of temperatures of a given species in their range, etc.

Another issue is the use of "Encephalization quotient" to refer to relative brain size. If I understood well, the authors use residuals from a log-log regression between brain size and body size (often called "Brain size residuals"), but instead they call it Encephalization quotient or EQ. This could be misleading as Encephalization quotient is calculated as Observed:Expected brain size whereas residuals are Observed - Expected (See Deaner et al. 2007). Although they give very similar results, they are not the same. Therefore, I suggest using the more general term "Relative brain size" in the text and then explain this was calculated as phylogenetically corrected residuals. Or if you prefer, use "Brain residuals" or "brain size residuals" in the tex.

Regarding the discussion, in the paragraph starting with line 402, I am not entirely convinced by the way it is framed. The authors suggest it is a matter of scale (i.e., some factors are only notable at certain scales), but it could be the case that some ecological/environmental factors are not relevant or there is no variation in the trait at certain lower level (i.e., social vs non-social cannot be compared in non-social groups), so in this sense, rather than taxonomic level per se, what matters is variation among traits. To be able to detect differences in correlations between brain size and other traits within families, you need sufficient variation in both brain size and predictors, which will be less likely as you look at smaller samples of species. I suggest rephrasing this paragraph a bit to discuss these ideas.

Finally, the very last lines of a manuscript are usually kept for conclusions and implications, but in this manuscript you talk about potential weaknesses (line 427 "In our sampling the body mass alone is responsible for more than 80% of brain" ... "this does not mean that research using EQ are not relevant") and limitations (line 430 "The encephalization quotient remains the only measurement accessible in many cases". I think this ending makes the reader wonder if all the results that have been presented before are relevant at all. I suggest rephrasing so that the last paragraph/sentences summarise the main conclusions of the study and why it is relevant for the broader field.

Specific comments

Line 38. "complex environments" is a very ambiguous term. Cognitive buffer hypothesis usually refers to changing environments (i.e., seasonal variation of food or colonization of a different environment) but I do not understand what you mean by "complex environments" here.

Line 167. Remove dot after parenthesis. In same line, Grey should be in lower case.

Line 172. The sentence "as expected following the allometric model" was confusing to me. I do not understand why is expected that this families will have a relatively smaller brain. Could you please clarify?

Line 191. I was surprised that you find this rate decrease only in 23% of the tree but it is still significant (line 189). Is this correct?

Line 245 (Figure 3). The figure caption ("Correlation between brain size evolution and our variables") is misleading as this is not including brain size but correlation between predictors, right?

Line 228-230. It is the first time that "Group size" and "Social Complexity" groups of analyses are mentioned so it could be confusing. I was only able to understand it when reading the methods. I suggest rephrasing and briefly introducing here the two types/groups of analysis and maybe refer to methods for more details.

Line 321. There is a recent study on body shape evolution on carnivores that might be relevant to cite here (Law 2021)

Line 374-376. This sentence is difficult to follow, I suggest rephrasing.

References

Sol, D., Stirling, D. G., & Lefebvre, L. (2005). Behavioral drive or behavioral inhibition in evolution: subspecific diversification in Holarctic passerines. *Evolution*, 59(12), 2669-2677.

Sayol, F., Lapiedra, O., Ducatez, S., & Sol, D. (2019). Larger brains spur species diversification in birds. *Evolution*, 73(10), 2085-2093.

Creighton, M. J., Greenberg, D. A., Reader, S. M., & Mooers, A. Ø. (2021). The role of behavioural flexibility in primate diversification. *Animal Behaviour*, 180, 269-290.

Deaner, R. O., Isler, K., Burkart, J., & Van Schaik, C. (2007). Overall brain size, and not encephalization quotient, best predicts cognitive ability across non-human primates. *Brain, behavior and evolution*, 70(2), 115-124.

Law, C. J. (2021). Ecological drivers of carnivoran body shape evolution. *The American Naturalist*, 198(3), 406-420.

Reviewer #2 (Remarks to the Author):

The authors seek to determine the effects of ecological, social, environmental, and physiological factors on brain size evolution across Carnivora and whether this differs among clades. To do this they have tested for differences in the evolutionary tempo of EQ as well as through correlations between EQ and variables associated with ecology, sociality, environment, and physiology.

Comments:

The manuscript is lacking a hypothesis statement and prediction, making it challenging to determine the goals of the manuscript and thus whether the analyses included within the study were appropriate. The authors also claim this is the first study to evaluate brain size in relation to a wide range of variables, but they fail to acknowledge (and cite) Gittleman 1986 "Carnivore brain size, behavioral ecology, and phylogeny", which tests for a relationship between relative brain size and the same variables included in this study across approximately 150 carnivoran species. The authors also test for differences in brain to body size allometry among carnivoran families and rates of evolution of EQ. These analyses have been tested by previous authors multiple times across mammals, including Carnivora (see Smaers 2012, 2019, 2021 and Burger et al 2019). It would, therefore, benefit the authors considerably to discuss how their study is unique from work that has been previously done.

As it currently stands, this study is not reproducible because some of the methods are lacking in detail. For example, the cranial measurements taken, though referenced, are never depicted or described and would benefit greatly from the inclusion of a figure. The variables being tested against EQ should be described in more detail as variables such as diet and activity pattern are a bit vague and can be defined in multiple ways. Specifically, when considering the Expensive-Tissue hypothesis both diet quality and dietary variation can play a role in brain size. The specimens measured also include an uncatalogued fossil, which should be excluded to ensure reproducibility. The authors also state that they ran analyses for all clades with more than 10 species included: Canidae, Felidae, Mustelidae. Their phylogeny, however, includes more than 10 species for Herpestidae and Viverridae. The authors should explain why these clades were not studied independently or correct their phylogeny to only include those taxa included in the study. I would also encourage the authors to submit their R code for review.

The authors use linear measurements of the cranium to estimate brain volume, and while this has been done by Finarelli 2006, the authors should justify why these measurements are more or equally appropriate over actual measures of brain volume obtained via beads or CT scan. The brain volume estimates calculated through the models provided by Finarelli 2006 contain error, which should in the least be discussed in the methods. The authors should note other studies that have used these measurements as well to further justify their use.

There is also a disconnect between the reported results and the discussion section. The authors discuss the significant relationships between caniforms, feliforms, Canidae, Felidae, and Mustelidae in the results, but then in the discussion talk extensively about Eupleridae, for whom no results were presented. To remedy this, the authors should include the results, even those that are not significant, for all the tested variables across all clades within the text. The authors also discuss the negative association between EQ across Carnivora and geographic range and how this can be interpreted. However, this correlation was not significant, making any further interpretation inappropriate. In addition, while correlations between brain size shifts and environment may correlate in mustelids, the authors misquote Law 2019, who stated that body length shifted prior to skull shape in evolutionary history, not EQ. This should be corrected.

The figures and tables should also be modified to be more easily interpreted. Figure 1, while containing informative components, has entirely too many things within it. It results in text that is too small to read and lines and points that are difficult to follow. It should be separated into multiple figures. I would also avoid light colors such as yellow. Figure 3 has a misleading title, as brain size is not a variable being tested in this figure. The asterisks should be defined to indicate what *, **, and *** represent. Supplementary Table 4 depicts the same information as Figure 3 and is more informative. I recommend replacing Figure 3 with Supplementary Table 4.

Supplementary Figure 1 is extremely informative, and the results of what I interpret to be the main purpose of this paper. This figure, or at the very least those relationships found to be significant, should be included in the main text.

The large size of the supplementary tables causes them to run over onto multiple pages. This in

turn means the column names and species associated with each row are lost and this should be corrected.

Reviewer #1 (Remarks to the Author):

This study focus on brain size evolution in Order Carnivora. The authors have compiled an impressive dataset of 174 species and show how the rate of brain size evolution have changed in different places of the phylogeny of carnivores. In addition, the authors also test which traits might be more influential for the evolution of brain size. The study is important and interesting, and the figures are very well presented (the reconstructed skulls are super nice!). However, I have several comments and concerns that might need to be addressed prior to publication.

Response : We would like to thank the reviewer for this kind comment.

A first general comment is the inclusion of certain variables in the analysis. This is a bit of a philosophical debate, but I am not comfortable with the inclusion of geographical range as a predictor of brain size. All the independent variables (predictors) included are either species traits (life-history, sociality, etc.) that could influence the fitness of other traits (here brain size) or abiotic/environmental variables (i.e., temperature) that one could also argue that can change the fitness of other traits (i.e., in this case brain size). However, geographical range is neither a “trait” nor a proper environmental variable that can be a selective agent (but maybe a proxy for some other variable?). In this sense, individuals of a species do not possess a “geographical range” value that could be inherited or selected. I can see why geographical range could be useful in other species comparison contexts (it can be a predictor of extinction risk or maybe a response of certain traits), but I do not see how it could affect brain size evolution. So, my suggestion would be to either exclude it from the analysis, or instead, explain which is the prediction of including this variable.

Response : We thank the reviewer for this very relevant comment. It is true that according to its definition, the geographic range is not a trait of an individual, nor an environmental parameter that can influence fitness. On the other hand, the geographic distribution is a good proxy for the ability of a species - and therefore of the individuals that composed it - to immigrate, survive and thrive in different environments, and traits which influence range limits are known to evolve under natural selection. Therefore, in our study we used the geographic range as an accessible proxy of species abilities to disperse and colonize new geographic landscapes. Under this definition, the geographic range is a variable that may affects fitness, and therefore can potentially influence the evolution of the relative brain size within these species. We recognize that, as you have pointed out, this is a bit of a philosophical debate, but we hope that we have been able to explain our choice as best as possible. We have included this information in the methodology section to emphasize this point and we remain open to discussion (L. 596 – 601).

Related to the previous comment, the significant effect of Geographical range is very difficult to interpret, as it could mean different things. For instance, the fact that large-brained species have smaller ranges is interpreted as a lower ability to colonize new habitats (Line 363). However, it might be the case that large-brained species are better able to colonize new habitats, and then speciate, giving rise to a lot of small-ranged species. Indeed, some previous studies have found that large-brained mammals and birds tend to speciate more often (Sol et al. 2005; Sayol et al. 2019; Creighton et al. 2021). Therefore, if geographical range is maintained in the analyses, I think that a more nuanced discussion could help to interpret its significance.

Response : This is indeed a very interesting hypothesis, and we thank the reviewer for raising this possibility. Under this hypothesis, species with a small relative brain size should have lower speciation rates compared to species with a large relative brain size. Similarly, one could assume a significant effect of the geographic range on the diversification rate which could lead to the macroevolutionary pattern we observe in our study. We thus decided to implement new analyses

to test this possibility. To assess the relationship between the diversification rate and the relative brain size as well as the effect on geographic range at the species level, we first estimated diversification rates (DR) for each terminal branch of the topology (i.e. each species included in our dataset). Then, we used phylogenetic generalized least squares (PGLS) regressions using DR as the response variable while the relative brain size, the geographic range, and the interaction between these two parameters were used as predictors. Our analyses revealed no significant influence of encephalization ($t = -0.48$; $p = 0.63$) or geographic range ($t = -0.68$, $p = 0.50$) on the rate of diversification within terrestrial carnivorans. Similarly, the interaction between the relative brain size and the geographic range did not significantly impact the rate of diversification for these species ($t = 0.43$, $p = 0.68$). Therefore, we can confidently exclude this hypothesis to explain the relation between the geographic range and the relative brain size evolution in Carnivora. We added a paragraph in the methodology (L. 663 – 676) and results section (L. 340 – 344) and discussed these results further on the revised version of the manuscript (L. 460 –464).

Regarding the inclusion of temperature. I guess the main reason is that previous works have found that environmental variation is a good predictor of relative brain size. However, note that it is “environmental variation” what matters and not temperature per se, so it might be interesting to include other environmental variables that directly measure variation such as amplitude or SD of temperatures of a given species in their range, etc.

Response : We recognize that the variation of the average temperature could be a very interesting data to analyse within the framework of our study. However, the database that we used to extract the environmental data does not allow access to this information (i.e. Pantheria). Similarly, we did not have access to the species distribution map computed by Sechrest (2003) used by the Pantheria database to extract the minimum/maximum temperature data and it does not seem relevant to us to mix several databases with different acquisition dates and techniques. Nevertheless, we believe that the average temperature is still quite important for the study of the mammalian brain evolution, notably because we found that the average temperature is one of the predictors that best estimate the relative brain size within Mustelidae and feliformian species. We added a paragraph in the discussion to emphasize the fact that it would be important to analyse the temperature variation in addition to the average temperature in future studies to draw finer conclusions (L. 478 – 487).

Another issue is the use of “Encephalization quotient” to refer to relative brain size. If I understood well, the authors use residuals from a log-log regression between brain size and body size (often called “Brain size residuals”), but instead they call it Encephalization quotient or EQ. This could be misleading as Encephalization quotient is calculated as Observed:Expected brain size whereas residuals are Observed – Expected (See Deaner et al. 2007). Although they give very similar results, they are not the same. Therefore, I suggest using the more general term “Relative brain size” in the text and then explain this was calculated as phylogenetically corrected residuals. Or if you prefer, use “Brain residuals” or “brain size residuals” in the text.

Response : Thank you for this relevant comment. Indeed, the measurement that we used corresponds to the relative brain size calculated as the residuals of the regression between the log of the body mass and the log of the estimation endocranial volume (i.e., Observed – Expected). We have corrected this in your manuscript to avoid confusion.

Regarding the discussion, in the paragraph starting with line 402, I am not entirely convinced by the way it is framed. The authors suggest it is a matter of scale (i.e., some factors are only notable at certain scales), but it could be the case that some ecological/environmental factors are not relevant or there

is no variation in the trait at certain lower level (i.e., social vs non-social cannot be compared in non-social groups), so in this sense, rather than taxonomic level per se, what matters is variation among traits. To be able to detect differences in correlations between brain size and other traits within families, you need sufficient variation in both brain size and predictors, which will be less likely as you look at smaller samples of species. I suggest rephrasing this paragraph a bit to discuss these ideas.

Response: This is a very relevant point. To explore this question, we decided to implement new analyses in our manuscript. Specifically, for the categorical predictors, we used a chi-squared analyses to test for proportions homogeneity of the different categories according to the taxonomic groups. Similarly, we performed Bartlett's test to assess the homogeneity of variances for continuous predictors at different taxonomic scales in parallel with ANOVA tests which provides information on the differences in means between these groups. All of these tests were performed between each taxonomic groups (i.e., entire order Carnivora, the suborders Caniformia and Feliformia, as well as families with more than 20 species in our sample, the Canidae, Felidae and Mustelidae families). All this information has been made available in a detailed Supplementary Data 3 which allows a quick visualization of the predictors' significant statistical differences depending on taxonomic groups and scales. We added a paragraph in the methodology (L. 618 – 628) and results sections (L. 340 – 344) regarding these analyses. In addition, we have modified part of our discussion to deal more specifically with this question (L. 517 – 531).

Finally, the very last lines of a manuscript are usually kept for conclusions and implications, but in this manuscript you talk about potential weaknesses (line 427 “In our sampling the body mass alone is responsible for more than 80% of brain” ... “this does not mean that research using EQ are not relevant”) and limitations (line 430 “The encephalization quotient remains the only measurement accessible in many cases”). I think this ending makes the reader wonder if all the results that have been presented before are relevant at all. I suggest rephrasing so that the last paragraph/sentences summarise the main conclusions of the study and why it is relevant for the broader field.

Response : We understand what the reviewer points out and agree with this comment. We have changed this part of the discussion to present the main conclusions of our study as well as potential avenues of research that would be interesting to explore in order to complete our discoveries (L. 541 – 549).

Specific comments

Line 38. “complex environments” is a very ambiguous term. Cognitive buffer hypothesis usually refers to changing environments (i.e., seasonal variation of food or colonization of a different environment) but I do not understand what you mean by “complex environments” here.

Response : We agree and changed “complex environments” by “changing environments”.

Line 167. Remove dot after parenthesis. In same line, Grey should be in lower case.

Response : We thank the reviewer for pointed out these mistakes.

Line 172. The sentence “as expected following the allometric model” was confusing to me. I do not understand why is expected that this families will have a relatively smaller brain. Could you please clarify?

Response : The “allometric model” (or standard allometric model) that we referred is a model that only include the body mass as a predictor for the brain size and for which all the observed variations of brain mass are explained by this predictor (here, the body mass). More concretely, this result can be transcribed visually by the slope obtained from the regression between the log of the body mass

and the log of the brain mass for the entire dataset (Figure 1A, black dotted slope). We have added elements to better explain this and hope that it is now clearer.

Line 191. I was surprised that you find this rate decrease only in 23% of the tree but it is still significant (line 189). Is this correct?

Response : This is indeed correct. The phylogenetic ridge regression approach we used highlighted a significant decrease in evolutionary rates for relative brain size evolution for this specific node. Nevertheless, the topology of the phylogenetic trees being subject to debate, we decided to test the impact of topological uncertainties on our results. It turns out that only 23% of the trees estimated by our methods (i.e. having different randomly generated topologies) show a significant decline in evolution rates. This means that the topological uncertainty has a strong influence for the estimation of evolution rate changes for this particular node.

Line 245 (Figure 3). The figure caption (“Correlation between brain size evolution and our variables”) is misleading as this is not including brain size but correlation between predictors, right?

Response : We thank the reviewer for finding this error. It is indeed the correlations between the different predictors used in our study. We have changed the title of the figure accordingly (now figure 5 in the new version of the manuscript).

Line 228-230. It is the first time that “Group size” and “Social Complexity” groups of analyses are mentioned so it could be confusing. I was only able to understand it when reading the methods. I suggest rephrasing and briefly introducing here the two types/groups of analysis and maybe refer to methods for more details.

Response : We understand that this point can be confusing due to the structure of the article, and we have added a paragraph to explain these different analyses in the results section (L. 108 – 112).

Line 321. There is a recent study on body shape evolution on carnivores that might be relevant to cite here (Law 2021)

Response : We thank the reviewer for proposing this very interesting citation for our manuscript. In particular, we have added a few words concerning the evolutionary shift in body elongation highlighted by Law (2021) within the Mustelidae family which could have a link with our results on the evolution of the relative brain size in this group (L. 396).

Line 374-376. This sentence is difficult to follow, I suggest rephrasing.

Response : We have rewritten this sentence and hope that it is now clearer.

References

Sol, D., Stirling, D. G., & Lefebvre, L. (2005). Behavioral drive or behavioral inhibition in evolution: subspecific diversification in Holarctic passerines. *Evolution*, 59(12), 2669-2677.

Sayol, F., Lapiedra, O., Ducatez, S., & Sol, D. (2019). Larger brains spur species diversification in birds. *Evolution*, 73(10), 2085-2093.

Creighton, M. J., Greenberg, D. A., Reader, S. M., & Mooers, A. Ø. (2021). The role of behavioural flexibility in primate diversification. *Animal Behaviour*, 180, 269-290.

Deaner, R. O., Isler, K., Burkart, J., & Van Schaik, C. (2007). Overall brain size, and not encephalization

quotient, best predicts cognitive ability across non-human primates. *Brain, behavior and evolution*, 70(2), 115-124.

Law, C. J. (2021). Ecological drivers of carnivoran body shape evolution. *The American Naturalist*, 198(3), 406-420.

Reviewer #2 (Remarks to the Author):

The authors seek to determine the effects of ecological, social, environmental, and physiological factors on brain size evolution across Carnivora and whether this differs among clades. To do this they have tested for differences in the evolutionary tempo of EQ as well as through correlations between EQ and variables associated with ecology, sociality, environment, and physiology.

Comments:

The manuscript is lacking a hypothesis statement and prediction, making it challenging to determine the goals of the manuscript and thus whether the analyses included within the study were appropriate. **Response : We added a sentence at the end of the introduction in order to clearly define the main objectives of our manuscript (L. 83 – 85). This study being mostly exploratory, with many predictors studied and several taxonomic groups, we decided not to add all the hypotheses that we had at the beginning of the study, which included among many others, the impact of the social environment, the type of habitat, the diet, the reproduction, and many other parameters which may differ according to the families. We are afraid that it may only blur the main objectives of the article.**

The authors also claim this is the first study to evaluate brain size in relation to a wide range of variables, but they fail to acknowledge (and cite) Gittleman 1986 “Carnivore brain size, behavioral ecology, and phylogeny”, which tests for a relationship between relative brain size and the same variables included in this study across approximately 150 carnivoran species. The authors also test for differences in brain to body size allometry among carnivoran families and rates of evolution of EQ. These analyses have been tested by previous authors multiple times across mammals, including Carnivora (see Smaers 2012, 2019, 2021 and Burger et al 2019). It would, therefore, benefit the authors considerably to discuss how their study is unique from work that has been previously done.

Response : The study of Gittleman (1986) was very important in the design of our study, and we regret that the reviewer found that we did not highlight this research enough. In particular, we cited this article from the introduction (citation number 54) to recall that important research has already been carried out on brain evolution within carnivorans. To better acknowledge the importance of this work, we have also added this quote at line 76. However, we believe that our study has provided new and unpublished element about brain evolution for this group, notably by the evaluation of new variables, the investigation of the tempo of brain size diversification using up-to-date methods, and the fact we carried out phylogenetic analyses at different taxonomic scales. Contrary to the articles quoted by the reviewers for which we recognize and cite they major interest in this field, the main objective of our manuscript is not to compare carnivoran encephalization patterns to other mammals, but to draw the most complete portrait possible of the evolution of the relative brain size within this taxonomic group. We have added some elements in the introduction to support our point and hope that the main objective of our work is now clearer.

As it currently stands, this study is not reproducible because some of the methods are lacking in detail. For example, the cranial measurements taken, though referenced, are never depicted or described and would benefit greatly from the inclusion of a figure.

Response : Following this advice we added a figure (Supplementary Figure 2) allowing the readers to visualize how the measurements were taken according to method proposed by Finarelli (2006). All the measurements we took in order to estimate the endocranial volume are available for each specimen in the Table S1 and allows the reproducibility of the analyses.

The variables being tested against EQ should be described in more detail as variables such as diet and activity pattern are a bit vague and can be defined in multiple ways.

Response : We agree with the reviewer and have added a supplementary document which presents all of these variables and the definition of each category (Supplementary Data 1).

Specifically, when considering the Expensive-Tissue hypothesis both diet quality and dietary variation can play a role in brain size.

Response : It is true that these variables would have been very interesting to consider to better understand brain size evolution at macroevolution scale. Nevertheless, we lack precise data concerning the quality and variation of most carnivoran species diet, which would require numerous observations in natural environments. As we do not have access to this information, we preferred to focus on diet categories, but we hope that these problematics would be addressed in future research for this taxonomic group. We have added a point in our discussion regarding this topic (L. 554 – 549).

The specimens measured also include an uncatalogued fossil, which should be excluded to ensure reproducibility.

Response : We do not have any fossil species in our sampling, but we believe the reviewer is referring to the extant spotted hyaena (*Crocota crocuta*) specimen that we acquired from the African Fossils database. This specimen does not indeed have a collection number, but it is available for download by anyone directly from the database and therefore in no way prevents the reproducibility of the study (<https://africanfossils.org/fauna/spotted-hyena-?o=1>). We have added the link allowing access to this specimen on the Supplementary Table 1.

The authors also state that they ran analyses for all clades with more than 10 species included: Canidae, Felidae, Mustelidae. Their phylogeny, however, includes more than 10 species for Herpestidae and Viverridae. The authors should explain why these clades were not studied independently or correct their phylogeny to only include those taxa included in the study.

Response : We thank the reviewer for pointing out this mistake. This is a typographical error as we had chosen to carry out analyses on families with only more than 20 species (i.e. Canidae, Felidae and Mustelidae families). We made the change in our manuscript.

I would also encourage the authors to submit their R code for review.

Response : Indeed, there seems to have been an error during the submission of the manuscript concerning the R code. We apologize for this inconvenience and will provide the R file during the revision.

The authors use linear measurements of the cranium to estimate brain volume, and while this has been done by Finarelli 2006, the authors should justify why these measurements are more or equally

appropriate over actual measures of brain volume obtained via beads or CT scan. The brain volume estimates calculated through the models provided by Finarelli 2006 contain error, which should in the least be discussed in the methods. The authors should note other studies that have used these measurements as well to further justify their use.

Response : We agree with the reviewer, and we have added several quotes from different articles based on this method as well as elements in our methodology to highlight the limitations of brain volume estimation through the use of cranial external measurements (L. 563 – 571).

There is also a disconnect between the reported results and the discussion section. The authors discuss the significant relationships between caniforms, feliforms, Canidae, Felidae, and Mustelidae in the results, but then in the discussion talk extensively about Eupleridae, for whom no results were presented. To remedy this, the authors should include the results, even those that are not significant, for all the tested variables across all clades within the text.

Response : The paragraph that discusses more specifically the case of the family Eupleridae focuses on two points: the relative brain size and the absence of evolutionary rate shift for this node in the topology which is often characteristic in island taxa. These results are presented in line 152 and 213 respectively. This paragraph does not discuss the influence of selected predictors on Malagasy carnivoran brain size evolution (a family that includes only 7 extant species) as PGLS analyses have only been carried out for taxonomic groups with more than 20 species.

The authors also discuss the negative association between EQ across Carnivora and geographic range and how this can be interpreted. However, this correlation was not significant, making any further interpretation inappropriate.

Response : The results of the PGLS analyses demonstrate that the geographic range is one of the variables that significantly impacts the evolution of relative brain size in carnivoran species (see Table 2A). More precisely, the model that best predict the evolution of the relative brain size in our sample is a model including both the home range (with a positive impact on relative brain size) and the geographic range (with a negative impact on the relative brain size). We believe that the reviewer may have been misled by the non-significant results of the linear correlation between the geographic range and the relative brain size presented in the Supplementary Figure 1. However, these results only examine one variable at a time and do not consider phylogenetic non-independence of species. Consequently, Supplementary Figure 1 must be considered as a useful element to better understand the distribution of our data, but contrary to the results of the PGLS analyses, in no case does it allow to conclude on the variables which together influence the relative brain size within Carnivora.

In addition, while correlations between brain size shifts and environment may correlate in mustelids, the authors misquote Law 2019, who stated that body length shifted prior to skull shape in evolutionary history, not EQ. This should be corrected.

Response : We thank the reviewer for finding this mistake. In this sentence, we meant that the shift in cranial shape identified by Law 2019 seemed to be a subsequent event to the diversification of the relative brain size revealed by our analyses. We have changed the sentence accordingly to avoid further confusion and hope that this part of the discussion is now clearer (L. 390 – 394).

The figures and tables should also be modified to be more easily interpreted. Figure 1, while containing informative components, has entirely too many things within it. It results in text that is too small to read and lines and points that are difficult to follow. It should be separated into multiple figures. I would also avoid light colors such as yellow.

Response : In order to allow a better reading of our figures, we followed the reviewer's recommendation and separated it into three new figures, enlarged the texts, and used a darker yellow for Herpestidae species.

Figure 3 has a misleading title, as brain size is not a variable being tested in this figure.

Response : We thank the reviewer for pointing out this error and we have changed the title of the figure.

The asterisks should be defined to indicate what *, **, and *** represent.

Response : We have added legend in tables and figures where necessary.

Supplementary Table 4 depicts the same information as Figure 3 and is more informative. I recommend replacing Figure 3 with Supplementary Table 4.

Response : We believe that a figure is often more appealing and easier to read than a table to illustrate a point. Nevertheless, we understand the reviewer's desire to add relevant information that is easily accessible to readers. We have therefore included the correlation values as well as the p-values of each analysis in the figure (now figure 5 in the new version of the manuscript).

Supplementary Figure 1 is extremely informative, and the results of what I interpret to be the main purpose of this paper. This figure, or at the very least those relationships found to be significant, should be included in the main text.

Response : As explained before, although this figure is informative to understand our data, it is not the main purpose of this paper. Unlike PGLS analyses, a simple linear correlation conducted on each variable separately and without considering the phylogenetic non-independence often results in misleading interpretations (see Dechmann & Safi, 2009; Wartel, 2019). Therefore, PGLS analyses are better fitted to analyse the correlated evolution of several predictors and endocranial volume and are consequently widely used in this context.

The large size of the supplementary tables causes them to run over onto multiple pages. This in turn means the column names and species associated with each row are lost and this should be corrected.

Response : We apologize for this inconvenience. We have uploaded the supplementary tables in the correct format for this revision.

Reviewers' comments:

Reviewer #2 (Remarks to the Author):

The authors have addressed many of my concerns and the manuscript is much stronger. The additional detail and clarification in the introduction really place the manuscript within the literature and make the goal of contributions clearer. The figures were separated appropriately, and the addition of tables are appreciated for interpretation of the study.

There are a few points where the manuscript could still use some clarification, however. Many of my previous comments of including results from or interpretation of results from Supplementary Figure 1 stem from a lack of description of those linear analyses anywhere in the manuscript or supplemental documents. I, therefore, as a reader assumed those results were PGLS regressions because phylogenetically corrected analyses are all that is discussed in the methods section. I recommend the addition of a methods section to the supplemental documents to ensure readers are clear in the type of analyses presented therein and the reasoning behind them. Based on the authors responses though, it may be more beneficial to conduct separate PGLS analyses for each variable rather than present results of linear regressions that the authors themselves state are inappropriate.

Since the initial submission of this manuscript, another manuscript addressing a very similar topic by Lynch and Allen 2022 (<https://doi.org/10.1159/000523787>) was published. This paper addresses many of the short comings presented by the authors within their own dataset, including quantification of dietary variation, geographic ranges, and climatic ranges such as temperature and precipitation. The authors should include this manuscript within their discussion and its implications toward the results of their study. It will likely re-shape sections of their introduction and discussion.

With these additional changes, this manuscript will make contributions to our understanding of carnivoran brain evolution.

Reviewer #2 (Remarks to the Author):

The authors have addressed many of my concerns and the manuscript is much stronger. The additional detail and clarification in the introduction really place the manuscript within the literature and make the goal of contributions clearer. The figures were separated appropriately, and the addition of tables are appreciated for interpretation of the study.

Response : We would like to thank the reviewer for their time and involvement in reviewing our manuscript.

There are a few points where the manuscript could still use some clarification, however. Many of my previous comments of including results from or interpretation of results from Supplementary Figure 1 stem from a lack of description of those linear analyses anywhere in the manuscript or supplemental documents. I, therefore, as a reader assumed those results were PGLS regressions because phylogenetically corrected analyses are all that is discussed in the methods section. I recommend the addition of a methods section to the supplemental documents to ensure readers are clear in the type of analyses presented therein and the reasoning behind them. Based on the authors responses though, it may be more beneficial to conduct separate PGLS analyses for each variable rather than present results of linear regressions that the authors themselves state are inappropriate.

Response : We agree with the reviewer and understand that the Supplementary Figure 1 could be misleading in the interpretation of our results. As recommended, we removed Supplementary Figure 1 from the manuscript and performed instead new PGLS analyses on each variable separately which are more appropriate to describe our data in a Supplementary Table 5. We have also added a paragraph in the method section to describe these analyses (L.690-693) .

Since the initial submission of this manuscript, another manuscript addressing a very similar topic by Lynch and Allen 2022 (<https://doi.org/10.1159/000523787>) was published. This paper addresses many of the shortcomings presented by the authors within their own dataset, including quantification of dietary variation, geographic ranges, and climatic ranges such as temperature and precipitation. The authors should include this manuscript within their discussion and its implications toward the results of their study. It will likely re-shape sections of their introduction and discussion. With these additional changes, this manuscript will make contributions to our understanding of carnivoran brain evolution.

Response : Indeed, we learned of this new publication after our first revision of the manuscript. We contacted the authors to inform them of our study and to propose future collaborations on carnivoran brain evolution. We added this article to our bibliography and discussed their results alongside our findings (L.76-78; L.475-488; L.506-508).